# Noncoding RNAs in Thyroid-Follicular-Cell-Derived Carcinomas

**DOI:** 10.3390/cancers14133079

**Published:** 2022-06-23

**Authors:** Marco De Martino, Francesco Esposito, Maria Capone, Pierlorenzo Pallante, Alfredo Fusco

**Affiliations:** 1Istituto per l’Endocrinologia e l’Oncologia Sperimentale (IEOS) “G. Salvatore”, Consiglio Nazionale delle Ricerche (CNR), Via S. Pansini 5, 80131 Napoli, Italy; marco.demartino2@unina.it (M.D.M.); francesco.esposito2@unina.it (F.E.); maria.capone12@studenti.unina.it (M.C.); 2Dipartimento di Medicina Molecolare e Biotecnologie Mediche (DMMBM), Università degli Studi di Napoli “Federico II”, Via S. Pansini 5, 80131 Napoli, Italy

**Keywords:** thyroid carcinoma, noncoding RNA, microRNA, long noncoding RNA, pseudogene

## Abstract

**Simple Summary:**

Thyroid tumors represent the most common neoplastic pathology of the endocrine system. Mutations occurring in oncogenes and tumor suppressor genes are responsible for thyroid carcinogenesis; however, the complete mutational landscape characterizing these neoplasias has not been completely unveiled. It has been established that only the 2% of the human genome codes for proteins, suggesting that the vast majority of the genome has regulatory capabilities, which, if altered, could account for the onset of cancer. Hence, many scientific efforts are currently focused on the characterization of the heterogeneous class of noncoding RNAs, which represent an abundant part of the transcribed noncoding genome. In this review, we mainly focus on the involvement of microRNAs, long noncoding RNAs, and pseudogenes in thyroid cancer. The determination of the diagnosis, prognosis, and treatment of thyroid cancers based on the evaluation of the noncoding RNA network could allow the implementation of a more personalized approach to fighting these pathologies.

**Abstract:**

Among the thyroid neoplasias originating from follicular cells, we can include well-differentiated carcinomas, papillary (PTC) and follicular (FTC) thyroid carcinomas, and the undifferentiated anaplastic (ATC) carcinomas. Several mutations in oncogenes and tumor suppressor genes have already been observed in these malignancies; however, we are still far from the comprehension of their full regulation-altered landscape. Even if only 2% of the human genome has the ability to code for proteins, most of the noncoding genome is transcribed, constituting the heterogeneous class of noncoding RNAs (ncRNAs), whose alterations are associated with the development of several human diseases, including cancer. Hence, many scientific efforts are currently focused on the elucidation of their biological role. In this review, we analyze the scientific literature regarding the involvement of microRNAs (miRNAs), long noncoding RNAs (lncRNAs), and pseudogenes in FTC, PTC, and ATC. Recent findings emphasized the role of lncRNAs in all steps of cancer progression. In particular, lncRNAs may control progression steps by regulating the expression of genes and miRNAs involved in cell proliferation, apoptosis, epithelial–mesenchymal transition, and metastatization. In conclusion, the determination of the diagnosis, prognosis, and treatment of cancer based on the evaluation of the ncRNA network could allow the implementation of a more personalized approach to fighting thyroid tumors.

## 1. Tumors Deriving from Human Thyroid Follicular Cells

Most thyroid neoplasias originate from genetic alterations of follicular cells. From these cells derive benign follicular adenomas (FTA) and thyroid carcinomas (TC), which can be further subdivided into well-differentiated (WDTC) and poorly differentiated (PDTC) ones. Among the former, we can include papillary (PTC, 80% of TCs) and follicular (FTC, 10% of TCs) carcinomas of the thyroid, while, among the latter, we can include PDTC and undifferentiated ones, also called anaplastic carcinomas (ATC, 2–5% of TCs) [1] (Figure 1).

Interestingly, a number of mutations in oncogenes and tumor suppressor genes responsible for the onset of thyroid tumors have been observed over the years. For example, rearrangements of the RET proto-oncogene (RET) with a series of molecular partners were reported in about 20% of PTCs, while neurotrophic receptor tyrosine kinase 1 (NTRK1) rearrangements were observed in 10% of PTCs [2,3]. In addition, point mutations of the B-Raf proto-oncogene (BRAF) were observed in about 40% of PTCs and in a fraction equal to 20% of ATCs [4]. Point mutations affecting the Ras proto-oncogene (RAS) and tumor protein P53 (TP53) genes are very common in the undifferentiated TCs [5,6]. Ras mutations, reported even in FTA, are also frequently detected in FTC, where fusions between the paired box 8 (PAX8) and peroxisome proliferator-activated receptor gamma (PPARG) genes are quite frequent [7] (Figure 1).

## 2. Noncoding RNAs (ncRNAs)

In the last few decades, it was established that only a small portion of the human genome (2%) has the ability to code for proteins, suggesting that the remainder of the genome has regulatory capabilities, which also accounts for the complexity of the human species [8]. However, most of the noncoding genome is transcribed, constituting the heterogeneous class of ncRNAs, whose alterations are associated with the development of several human pathologies, including cancer [8,9,10,11]. Hence, many scientific efforts have recently been focused on the characterization and elucidation of their biological role [8].

On the basis of their length, ncRNA transcripts are classified as long noncoding RNAs (lncRNAs, more than 200 bp), including pseudogenes, and small noncoding RNAs (sncRNAs, less than 200 bp), with the latter class comprising small interfering RNAs (siRNAs), microRNAs (miRNAs), small nuclear and nucleolar RNAs (snRNAs and snoRNAs), circular RNAs (circRNAs), transfer RNAs (tRNAs), and PIWI-interacting RNAs (piRNAs) [8,9,10,11,12]. In this review, we mainly focus on the involvement of miRNAs, lncRNAs, and pseudogenes in TC (Table 1).

### 2.1. MicroRNAs (miRNAs)

MiRNAs are small noncoding transcripts generally ranging from 18 to 25 nucleotides. They are single-stranded RNA molecules and are highly conserved at the evolutionary level. Their main function is to bind to the 3′-UTR of the mRNA of genes coding for proteins, inducing repression of the translation or degradation of the messenger molecule [13,14,15]. Their peculiar mechanisms of action allow them to directly modulate a series of cellular biological processes such as cell proliferation, differentiation, and death. Therefore, their deregulation is often associated with the development of several human diseases, including cancer [16]. In particular, different molecular signatures have been found associated with different tumor histotypes, representing important tools for differential diagnosis and prediction of prognosis in human cancer [17].

### 2.2. Long Noncoding RNAs (lncRNAs)

A series of noncoding transcripts, very heterogeneous in length, ranging from 200 to 100,000 nucleotides long, belong to the category of lncRNAs. Members of this class can be defined as sense, antisense, intronic, and intergenic lncRNAs, retrotransposons, and pseudogenes [18]. They are able to interact with both DNA and RNA molecules, as well as with proteins, regulating gene expression through a series of different mechanisms that include chromatin modifications, alternative splicing, and protein relocation inside the cell [19]. As previously reported for miRNAs, lncRNAs also control several important biological processes [20,21], and their deregulation is often associated with human cancer, acting as oncogenes or tumor suppressors [10,11,22].

### 2.3. Pseudogenes

Pseudogenes are DNA sequences that, despite resembling functional genes, have lost the ability to code for proteins due to the excessive accumulation of inactivating mutations [23]. They are conserved in the course of evolution and have been found in organisms of different biological origins [24]. They are generally classified, according to their origin, as processed pseudogenes (deriving from retrotransposition events), duplicated pseudogenes (deriving from the duplication and subsequent inactivation of functional genes), and disabled pseudogenes (deriving from the inactivation of pre-existing functional genes) [24,25].

For a long time, it was erroneously believed that they were nonfunctional entities, but it was very recently shown that they have key roles in various cell activities, such as cell growth and invasion, and their deregulation has been described in several cancers, including TC development [26,27,28]. Indeed, it has been reported that pseudogenes carry in their sequence elements responsive to miRNAs, and this peculiarity allows them to indirectly regulate the expression of the corresponding genes, subtracting a significant number of miRNAs from the cell pool and then behaving as competitive endogenous RNAs (ceRNAs) [29,30,31].

### 2.4. Natural Antisense Transcripts (NATs)

NATs are generated following the transcription of the opposite strand of genes coding for proteins. Thanks to the correspondence of sequence in the antisense orientation, they are able to modulate the expression levels of the cognate protein-coding gene through different mechanisms [32]. In fact, in the majority of cases, the base pairing between cis-NAT and cognate transcripts is responsible for the decreased expression of the complementary gene. Moreover, NATs may contribute to epigenetic modifications such as DNA methylation and post-translational modifications of core histones. Furthermore, it is hypothesized that, during cis-NAT transcription, the transcriptional complex is assembled at the promoter site of the cognate gene [32].

A NAT named COMET (correlated to MET) was found to be highly expressed in PTC samples harboring the V600E mutation in the BRAF gene or the RET/PTC oncogene. Its expression is highly correlated with the expression of the MET proto-oncogene (MET), previously described as overexpressed in most PTCs. Interestingly, the silencing of COMET resulted in the reduced expression of MET and the genes involved in the mitogen-activated protein kinase (MAPK) pathway, and it significantly reduced the motility and invasiveness of TC cells, validating the potential role of COMET overexpression in the development of PTC [33].

### 2.5. Circular RNAs (circRNAs)

CircRNAs are noncoding single-stranded covalently closed circular RNA, mainly produced by reverse-splicing of the exons of precursor mRNAs (pre-mRNAs). Since they do not have 5′ or 3′ free ends, they are very resistant to cutting by exonucleases; therefore, they are more stable with respect to the other linear RNA molecules in the cell. Even though some circRNAs have been shown to code for proteins, it seems that their main function is related to gene regulation. Indeed, many studies have shown that circRNAs act as a sponge for miRNAs, thereby inhibiting their function [34,35]. However, additional functions have been taken into consideration, such as (a) interaction with RNA-binding proteins (RBPs) and RNAs to form RNA–protein complexes, (b) protein synthesis, and (c) the transportation of miRNAs. Nevertheless, the biological function of most circRNAs is still unclear. Several studies have associated circRNA deregulated expression with several diseases, including cancer [36].

Several research groups have analyzed the expression of circRNAs in TCs of different histotypes and correlated their expression with the patient’s clinicopathological features, such as tumor size, locoregional and distant metastases, prognosis, and mortality. Recent reviews [34,35,36,37] reported these major results. Interestingly, the high stability of circRNAs and their possible detection in blood and other bodily fluids, such as saliva and urine, open the possibility of their evaluation for early and differential diagnosis of thyroid diseases. Among the circRNAs involved in TC, it is worth mentioning hsa_circ_0058124 that shows oncogenic activity promoting PTC cell proliferation, tumorigenicity, tumor invasion, and metastasis by modulating miRNA-218-5p, which targets the NUMB endocytic adaptor protein (NUMB) gene, whose expression is critical for the fulfillment of asymmetric division in stem cells. Consistently, high hsa_circ_0058124 expression is associated with a poor outcome in PTC patients [38].

However, for a detailed description of the circRNAs altered in TC, we refer readers to recent reviews focused on this topic [34,35,36,37,39].

## 3. Deregulation of miRNAs in TC

As far as the deregulated expression of miRNAs in TCs is concerned, several studies have reported that their aberrant expression is associated with each particular tumor histotype. A description of all the differentially expressed miRNAs in TCs with respect to healthy tissue would be beyond the scope of this review. Therefore, in this section, we examine only a few representative miRNAs that are specifically deregulated in PTC, FTC, and ATC or nonspecifically deregulated in all these neoplasias (Figure 2). For a fairly complete picture of miRNAs differentially expressed in the different histotypes of human TC, we refer readers to other reviews [40,41].

### 3.1. MiRNAs Deregulated in PTC

PTCs are the most frequent histotype of TC deriving from follicular cells; therefore, numerous studies concerning the aberrant expression of miRNAs have been carried out on this type of tumor. In addition to a series of miRNAs aberrantly expressed and sharing the anaplastic histotype, a series of miRNAs appear to be specifically deregulated in PTCs [40,41]. Among the miRNAs specifically upregulated in PTCs, we have miR-181a/c, let7b-5p, miR-10a-5p, miR-31, miR-122a, miR-551b-3p, miR-145, miR-181b, miR-182, miR-25-3p, miR-451a, miR-140-3p, and let7i (Figure 2). On the other hand, among the miRNAs specifically downregulated in PTCs, we have miR-204, miR-654-3p, miR-146a-5p, miR-219-5p, miR-451, and miR-199a-1 (Figure 2).

Several studies carried out on PTCs have shown that miR-181b is specifically overexpressed in this tumor histotype [42,43,44]. It was one of the first miRNAs to be described as increased in PTCs, while its expression was almost completely absent in normal thyroid (NT) tissue [44]. Interestingly, it was subsequently shown that the expression of this miRNA enhances the proliferation of tumor cells through the repression of the CBX7 protein, which acts as a tumor suppressor [45,46].

MiR-204 is downregulated in PTCs, suggesting a negative role in the control of cell growth and invasiveness. In fact, it is underexpressed in PTCs compared to normal tissue [47], and its downregulation is also correlated with the ability of PTCs to metastasize to the lymph nodes [48]. Its role as a tumor suppressor appears to be supported by functional experiments showing its ability to suppress proliferation and induce apoptosis in PTC-derived cell lines and its ability to block tumorigenicity in vivo [49].

MiR-654-3p, which is decreased in PTCs, also appears to play a role in tumor suppression, as shown in a recent study demonstrating that the downregulation of the levels of this miRNA favors epithelial–mesenchymal transition (EMT), increases cell proliferation, and blocks apoptosis [50,51].

### 3.2. MiRNAs Deregulated in ATC

Although fewer studies have been performed, over the years, on ATCs than on PTCs, a series of aberrantly expressed miRNAs in this tumor histotype have nevertheless been highlighted. In this regard, it is interesting to note that most of the miRNAs selectively deregulated only in ATCs show purely tumor suppressor behavior, thus demonstrating their role in tumor progression toward the acquisition of a highly aggressive phenotype [40,41]. Among the miRNAs overexpressed exclusively in ATCs, we can observe miR-214, miR-302, and the miR-17-92 cluster (Figure 2). Instead, among the miRNAs exclusively downregulated in ATCs, we have miR-25, miR-26a, miR-125, miR-29, miR-125b, and miR-200 (Figure 2).

In particular, it has been reported that miR-30a and miR-200 are downregulated in ATCs compared to NT tissue, as well as compared to other WDTCs, thus behaving as tumor suppressor genes [52,53]. The downregulation of miR-30a, in particular, has been associated with an increase in tumor aggressiveness with a marked loss of differentiation, and with a final decrease in overall survival [52]. This behavior is explained by the ability of miR-30a to negatively regulate EMT, specifically migration and invasion, by inhibiting the expression of EMT markers [52].

Moreover, miR-200 and other members of this family are involved in the repression of EMT, with their loss being associated with an increase in tumor invasiveness [54,55]. Interestingly, members of the miR-200 family negatively regulate the expression of EMT markers such as zinc finger E-box-binding homeobox (ZEB) 1 and 2, Snail family transcriptional repressor 2 (SNAI2), and transforming growth factor beta (TGFβ) 2, thereby strengthening the maintenance of the epithelial phenotype and repressing the ability of cancer cells to metastasize [54,55,56].

MiR-125b is also underexpressed in ATCs and behaves as a tumor suppressor. In fact, following the restoration of its expression in cultured cells derived from ATC, cell proliferation was drastically repressed, thus supporting an important role of its downregulation in thyroid tumorigenesis [57]. It was subsequently shown that miR-125b is able to repress cell proliferation by arresting cells in the G1/S phase via the repression of cyclin-dependent kinase 6 (CDK6) and CDC25 [58] and by increasing cyclin-dependent kinase inhibitor 1A (CDKN1A) [59]. Its involvement in human carcinogenesis is further supported by the observation of its downregulation in non-TCs [60].

However, miRNAs deregulated in the anaplastic histotype are not necessarily tumor suppressors but sometimes behave similarly to real oncogenes. This is the case for members of the miR-17-92 cluster, which were found to be overexpressed in ATCs, thus conferring a growth advantage to the tumor [61]. It is interesting to note that functional experiments have shown that the individual silencing of miR-17-3p, miR-17-5p, and miR-19a suppresses cell proliferation, while it also induces apoptosis in the case of miR-17-3p [61]. Conversely, the forced expression of miR-17-5p and miR-19a is able to repress the expression of RB transcriptional corepressor 1 (RB1) and phosphatase and tensin homolog (PTEN), respectively, thus explaining their ability to enhance the proliferative capacity of tumor cells [61].

### 3.3. MiRNAs Deregulated in Both PTC and ATC

A series of miRNAs have been found deregulated in both PTCs and ATCs, undoubtedly indicating their basic role in contributing to thyroid carcinogenesis [40,41]. Among them, miR-221, miR-222, miR-155, miR-187, miR-224, miR-205, and miR-146b (Figure 2), which are overexpressed, are likely the most important.

Several studies have shown that miR-146b is highly overexpressed in PTCs [62,63,64,65]. Therefore, it is not surprising that it has also been proposed as a prognostic biomarker, given its strong association with a poor prognosis [65]. Furthermore, its high expression has also been associated with the presence of lymph node metastases in PTCs [48]. MiR-146b has a critical role in controlling important cellular functions such as proliferation, migration, invasion, and resistance to treatments [65,66,67]. Functional experiments in PTC-derived cell lines have shown that miR-146b is able to target SMAD family member 4 (SMAD4), whereas the inhibition of miR-146b allows its expression. SMAD4 is an actor of TGFβ signaling; therefore, the inhibition of miR-146b allows obtaining a more effective response in terms of the reduction in proliferation following the antiproliferative signaling of TGFβ [68]. It is interesting to note that miR-146b is also overexpressed in ATCs, thus highlighting a fundamental role in thyroid tumorigenesis starting from the initial stages up to the acquisition of a completely aggressive phenotype.

The miR-221/222 cluster, as already observed for miR-146b, is overexpressed in both PTCs and ATCs, thus underlining a role that occurs from the onset of the tumor to its subsequent development [15]. The overexpression of this cluster has been shown by different research groups in PTCs [42,43,44,69], and the evaluation of their expression has been proposed as an independent risk factor for the evaluation of recurrence in PTCs [70]. One of the mechanisms via which miR-221/222 acts is by targeting the messenger RNA of CDKN1B (cyclin-dependent kinase inhibitor 1B), facilitating the consequent entry of TC cells into the S phase [57,71].

### 3.4. MiRNAs Deregulated in FTC

Despite covering a low percentage of tumors generated by thyroid follicular cells, FTCs, which represent another malignant well-differentiated histotype, have attracted substantial attention regarding their study and characterization, as they show a different tumorigenic pathway when compared to PTCs, and their distinction from benign FTAs represents a major problem in the diagnosis of neoplastic thyroid diseases [72]. However, a molecular expression signature that differentiates FTCs from FTAs has not yet been detected. Therefore, the evaluation of miRNAs deregulated in FTCs can contribute to the establishment of a correct differential diagnosis in clinical practice. Studies have reported that a series of miRNAs are almost exclusively deregulated in FTCs compared to their nonpathological thyroid counterpart. Upregulated miRNAs include miR-192, miR-197, miR-328, miR-346, miR-10b, miR-92a, miR-107, miR-221-3p, miR-137, miR-767-5p, miR-663b, miR-30d-3p, miR-874-3p, miR-125a-3p, miR-1287-5p, miR-30c-1-3p, and miR-125b-2-3p (Figure 2) [41]. MiRNAs downregulated specifically in FTCs include miR-9-5p, miR-9-3p, miR-205-5p, miR-34b-3p, miR-1, miR-130b-5p, miR-328-3p, miR-10a-5p, miR-296-5p, and miR-138-5p (Figure 2) [41].

In particular, it has been observed that the levels of miR-197 and miR-346 are extremely high in FTCs when compared with FTAs, highlighting the potential of their detection for a differential diagnosis between FTC and FTA. Moreover, functional experiments performed on FTC-derived cells (FTC133 and K5) showed that these two miRNAs are directly involved in the control of cell proliferation [73]. Further studies have demonstrated that miR-197 and miR-346 modulate cell growth by acting directly on their target genes, i.e., activin A receptor type 1 (ACVR1) and tetraspanin 3 (TSPAN3) for miR-197 and EGF containing fibulin extracellular matrix protein 2 (EFEMP2) for miR-346 [73]. Another interesting study showed that miR-10b and miR-92a are extremely overexpressed in metastatic minimally invasive FTC (MI-FTC) and widely invasive FTC (WI-FTC), thus suggesting that these two miRNAs may play a key role in inducing the metastatic potential in the first steps of MI-FTC, thus making them highly aggressive [74]. Consistently, miR-10b and miR-92a are involved in the control of cell migration and invasion by acting directly on some gene targets, such as E-cadherin in the case of miR-92a [75].

However, it is interesting to note that a number of miRNAs are also simultaneously deregulated in PTCs and ATCs. In fact, miR-146-3p and miR-181 are overexpressed in both FTC and PTC, while miR-137 is overexpressed in both FTC and ATC (Figure 2). It is also noteworthy that miR-146b, miR-221, miR-222, miR-155, miR-187, and miR-224 are simultaneously overexpressed in FTC, PTC, and ATC (Figure 2) [41]. On the other hand, miR-486, miR-138, miR-199b-5p, miR-7, and miR-144 are downregulated in both FTC and PTC, while only miR-30a is downregulated in both FTC and ATC (Figure 2) [41]. It is interesting to note that miR-199b-5p and miR-144 are downregulated in TCs, with their loss already occurring in the transition from normal thyroid epithelium to adenoma [76]. Hence, they could play a real role in the malignant transformation of the thyroid cell and, accordingly, could be used as potential prognostic biomarkers.

## 4. Deregulation of lncRNAs in TC

### 4.1. LncRNAs Deregulated in PTC

#### 4.1.1. Metallophosphoesterase Domain-Containing 2 (MPPED2) Antisense RNA 1 (AS1)

MPPED2-AS1, previously known as RP5-1024C24.1, represents one of the best-studied lncRNAs in TC. Indeed, according to an lncRNA expression profile performed on 12 PTCs with respect to four NT tissues (not matching with the cancer tissue of the 12 PTCs), about 600 up- and more than 2000 downregulated lncRNAs were found with a fold change >2, while six up- and eight downregulated lncRNAs were found with a fold change >10 [77]. Among the latter, the authors focused on characterizing the role of MPPED2-AS1 (Figure 3A).

This lncRNA is located on chromosome 11, in the antisense position with respect to the MPPED2 gene that encodes a metallophosphodiesterase protein, and it has been already reported as a candidate tumor suppressor in cervical cancer [78], neuroblastoma [79], glioblastoma [80], and oral squamous cell carcinoma [81]. Both these genes are downregulated in differentiated and undifferentiated carcinomas, and even in benign adenomas, with a statically significant positive correlation between them. This result suggested a coregulation of these genes, which was confirmed by the upregulation of MPPED2 induced in TC cells in which MPPED2-AS1 expression was restored by transfection [77]. Notably, the downregulation of MPPED2-AS1 and MPPED2 in benign FTA suggests a role of this event in the early steps of thyroid tumorigenesis.

Interestingly, the mechanisms via which MPPED2-AS1 positively regulates the expression of the MPPED2 gene come from studies performed on breast cancer; however, we can assume that the same mechanisms also act in TC. Indeed, it has been demonstrated that MPPED2 downregulation is associated with hypermethylation of its promoter region. Consequently, the treatment of breast cancer cells with a demethylating agent, such as azacytidine, induces the expression of the MPPED2 gene [82]. Intriguingly, MPPED2-AS1 interacts with DNA methyltransferase 1 (DNMT1), inhibiting its activity, which then reduces the hypermethylation status of MPPED2, thereby eliciting its expression [82] (Figure 3A). The inability of MPPED2 to induce MPPED2-AS1 expression further supports this mechanism.

The contribution of MPPED2-AS1 downregulation to TC development has been validated by the inhibitory effects of its overexpression exerted on proliferation and migration in the PTC-derived cell line TPC-1. Indeed, cell cycle analysis of MPPED2-AS1-overexpressing cells indicated that MPPED2-AS1 is able to affect cell proliferation by inhibiting the G1/S transition, with an increased cell number in the G1 phase with respect to the control-transfected cells [77]. The authors claimed that the effects of MPPED2-AS1 on cell proliferation and migration are mediated, at least in part, by the modulation of the phosphatidylinositol-4,5-bisphosphate 3-kinase/AKT serine/threonine kinase (PI3K/AKT) pathway. Indeed, MPPED2-AS1 overexpression increases PTEN levels, consequently leading to the dephosphorylation of Ser 73 in AKT [77] (Figure 3A).

Interestingly, very similar results were obtained when the same TC cells were transfected with MPPED2, thus validating the hypothesis that MPPED2-AS1 tumor suppressor activity occurs through the induction of MPPED2 expression by reducing its hypermethylation status [77]. However, it cannot be excluded that MPPED2-AS1 may also act through the PI3K/AKT pathway via an MPPED2-independent mechanism. There has so far been no report of the effects of MPPED2 on the PI3K/AKT pathway.

It is noteworthy that the downregulation of MPPED2-AS1 and MPPED2 does not seem to be restricted to thyroid neoplasms but seems to be a more general event in carcinogenesis. Indeed, a significant reduction in MPPED2-AS1 expression associated with that of MPPED2 has also been shown in breast cancer and glioblastoma, where it is also correlated with a poor prognosis [80,82]. From these cancer models, more information on its mechanisms was revealed, through which it regulates cell proliferation and migration. In fact, a significant reduction in cyclin D and E has been reported in MPPED2-transfected breast cancer cells, accounting for its inhibitory effect on G1/S transition [82]. Moreover, the overexpression of MPPED2 decreases ZEB1 levels and increases E-cadherin expression, accounting for the cell invasiveness effect [82] (Figure 3A).

However, the mechanisms via which MPPED2-AS1 and MPPED2 are involved in cell physiology and cancer are not completely unveiled. Certainly, the generation of null mice for MPPED2-AS1 and MPPED2 would give more insights into the mechanism of action of these two genes and would lead to the identification of other mechanisms via which their downregulation would contribute to the development of thyroid and non-thyroid neoplasias.

#### 4.1.2. The lncRNA X Inactive Specific Transcript (XIST)

The lncRNA XIST is a key factor in the regulation of X chromosome inactivation in mammals. Its overexpression was recently reported in a large variety of human malignancies in comparison with adjacent normal tissues [83]. Therefore, XIST can be regarded as a novel oncogene that acts by interacting with different miRNAs. Indeed, XIST suppression decreases cell growth and invasion and induces apoptosis in non-small-cell lung carcinoma (NSCLC) through the inhibition of miR-186-5p [83], while it promotes the progression of human hepatocellular carcinoma through the repression of the PTEN suppressor gene. It is noteworthy that, in esophageal squamous cell carcinoma (ESCC), XIST leads to increased levels of the enhancer of zeste homolog 2 (EZH2) by sequestering miR-101 [83].

Recently, XIST overexpression was also reported in PTC [84,85,86], where its higher levels are correlated with larger tumor size and advanced TNM stages. Moreover, overall survival is significantly lower in patients expressing high XIST levels [84]. Functional studies have also been performed to validate the potential role of XIST overexpression in thyroid carcinogenesis. One study [86] reported that XIST promotes the migration and invasion of PTC cells by sponging miR-101-3p, which, in turn, targets the CLDN1 gene that codes for claudin-1, a tight junction protein associated with cancer cell migration and invasion (Figure 3B) [86]. Another study [84] demonstrated that XIST depletion leads to a decreased cancer cell proliferation associated with a reduction in MET protein levels through sponging miR-34a, which is able to inhibit MET protein synthesis. Moreover, MET inhibition also results in the attenuation of PI3K and AKT mitogenic signaling, further supporting the role of XIST in the development of PTC (Figure 3B) [84]. Interestingly, the role of XIST in PTC cell proliferation, migration, and invasion was also reported and ascribed to the ability of XIST to target miR-141 [85]. Through sponging miR-101 (as shown in ESCC), it is also likely that XIST overexpression may increase the levels of EZH2, a protein that plays a critical role in ATC [87] and whose overexpression in this malignancy is supported by epigenetic mechanisms involving miRNAs and lncRNAs, as previously reported [83,88].

#### 4.1.3. The HOX Transcript Antisense RNA (HOTAIR) lncRNA

The lncRNA HOTAIR was found upregulated in human TC and inversely correlated with miR-1, whose inhibition leads to activation of cyclin D2 (CCND2), a member of the cyclin family, frequently misregulated in human cancer [89]. Interestingly, HOTAIR overexpression is associated with a poor outcome, showing high levels in the serum of patients with lymph node metastases of PTC [90]. Consistently, functional studies demonstrate that HOTAIR regulates EMT mediated by the WNT family member/β-catenin (WNT/β-catenin) pathway [91]. Therefore, HOTAIR may also be proposed as a molecular target for the treatment of lymph node metastases deriving from PTC.

#### 4.1.4. RP11-230G5.2

Among the upregulated lncRNAs, it is worthwhile to mention RP11-230G5.2 [77], located on chromosome 12, which represents the natural antisense transcript of the methionine sulfoxide reductase B3 (MSRB3) gene. It has been reported that high expression levels of MSRB3 in gastric carcinomas can predict peritoneal metastasis, recurrence, and a poor prognosis [92]. Furthermore, MSRB3 is involved in the regulation of proliferation and migration of gastric carcinoma cells. Consistently, MSRB3 is also upregulated in lung adenocarcinoma [93].

#### 4.1.5. AC079630.2

Another interesting lncRNA, upregulated in PTC, is represented by AC079630.2 [77], located on chromosome 12, whose expression is positively correlated with that of the leucine-rich repeat kinase 2 (LRRK2) gene. Interestingly, LRRK2 has been reported to play an important role in TC [94]. Indeed, silencing experiments in TC cell lines have shown that the downregulation of LRRK2 induces cell cycle arrest, promotes apoptosis, and inhibits cell migration through the inhibition of the C-Jun N-terminal kinase 1 (JNK) signaling pathway [94]. This observation suggests a possible contribution of the upregulation of AC079630.2 and LRRK2 to the development of PTC. Additionally, it was also recently reported in PTC that the overexpression of LRKK2 is associated with that of the lncRNAs RP11-476D10.1 [95] and H19 [96]. This observation strongly reinforces the idea that the LRRK2 gene plays a role in thyroid carcinogenesis.

**Table 1 cancers-14-03079-t001:** Summary of the main altered long noncoding RNAs and pseudogenes in thyroid carcinomas.

LncRNA Pseudogene	Alt. ^1^	Mechanism	Effect	Ca. Type ^2^	Ref. ^3^
MPPED2-AS1	↓	↔ DNMT1 → ↓ MPPED2	Proliferation	PTC	[77]
		↑ PI3K/AKT	Migration		
			Invasiveness		
XIST	↑	↓ miR-101-3p → ↑ CLDN1	Proliferation	PTC	[84,86]
		↓ miR-34a → ↑ MET → ↑ PI3K/AKT	Migration		
			Invasiveness		
HOTAIR	↑	↓ miR-1 → ↑ CCND2	Proliferation	PTC	[89,91]
		↑ WNT/β-catenin	Cancer progression		
RP11-230G5.2	↑	↑ MSRB3	Proliferation	PTC	[77]
			Migration		
AC079630.2	↑	↑ LRRK2	Proliferation	PTC	[77]
			Apoptosis		
PAR5	↓	↔ EZH2 → ↓ E-cadherin	Proliferation	ATC	[97]
			Migration		
			Stemness		
RMST	↓	↑ SOX2	Proliferation	ATC	[97] ^4^
			Migration		
			Stemness		
PTCSC3	↓	↑ miR-574-5p	Proliferation	PTC	[98,99]
		↑ STAT3 → ↑ INO80	Apoptosis	ATC	
			Drug resistance		
MALAT1	↑	↓ miR-363-3p → ↑ MCL1	Proliferation	PTC	[100,101,102]
		↑ WNT/β-catenin ↑ → CCND1	Apoptosis	ATC	
UCA1	↑	↓ miR-204 → ↑ BRD4	Proliferation	PTC	[103,104,105]
		↓ miR-135 → ↑ c-MYC		ATC	
H19	↑	↓ miR-17-5p → ↑ YES1	Proliferation	PTC	[106,107,108]
				ATC	
NEAT1	↑	↓ miR-214	Proliferation	PTC	[109,110]
		↓ miR-9-5p → ↑ SPAG9	Drug resistance	ATC	
PVT1	↑	↔ EZH2 → ↑ TSHR	Proliferation	FTC	[111]
GAS5	↓	↑ miR-221-3p → ↓ CDKN2B	Proliferation	FTC	[112]
BRAFP1	↑	↑ MAPK	Proliferation	PTC	[113]
DUXAP8	↑	↓ miR-20b-5p → ↑ SOS1 → ↑ KRAS	Proliferation	PTC	[114]
			Apoptosis		
			Invasiveness		
DUXAP10	↑	↑ AKT/mTOR	Proliferation	PTC	[115]
		↑ MMP-2	Apoptosis		
		↑ MMP-9	Invasiveness		
		↓ Caspase 3			
EGFEM1P	↑	Not known	Not known	PTC	[77]
HMGA1P6	↑	↑ HMGA1	Cancer progression	ATC	[116,117]
		↑ HMGA2			
		↑ EZH2			
HMGA1P7	↑	↑ HMGA1	Cancer progression	ATC	[116,117]
		↑ HMGA2			
		↑ EZH2			

^1^ Alteration; ^2^ cancer type; ^3^ references; ^4^ De Martino, M. et al. (manuscript in preparation). Abbreviations: FTC, follicular thyroid carcinoma; PTC, papillary thyroid carcinoma; ATC, anaplastic thyroid carcinoma; MPPED2-AS1, MPPED2 antisense RNA 1; XIST, X inactive specific transcript; HOTAIR, HOX transcript antisense RNA; PAR5, Prader-Willi/Angelman region RNA 5; RMST, rhabdomyosarcoma 2-associated transcript; PTCSC3, papillary thyroid carcinoma susceptibility candidate 3; MALAT1, metastasis-associated lung adenocarcinoma transcript 1; UCA1, urothelial cancer-associated 1; H19, H19 imprinted maternally expressed transcript; NEAT1, nuclear paraspeckle assembly transcript 1; PVT1, PVT1 oncogene; GAS5, growth arrest-specific 5; BRAFP1, BRAF pseudogene 1; DUXAP, double homeobox A pseudogene; EGFEM1P, EGF-like and EMI domain-containing 1 pseudogene; HMGA1P, high-mobility group A1 pseudogene; MPPED2, metallophosphoesterase domain-containing 2; PI3K, phosphatidylinositol-4,5-bisphosphate 3-kinase; AKT, AKT serine/threonine kinase; DNMT1, DNA methyltransferase 1; CLDN1, claudin-1; MET, MET proto-oncogene; CCND, cyclin D; WNT, WNT family member; MSRB3, methionine sulfoxide reductase B3; LRRK2, leucine-rich repeat kinase 2; EZH2, enhancer of zeste homolog 2; SOX2, SRY-box transcription factor 2; STAT3, signal transducer and activator of transcription 3; INO80, INO80 complex ATPase subunit; MCL1, MCL1 apoptosis regulator; BRD4, bromodomain-containing 4; MYC, MYC proto-oncogene; YES1, YES proto-oncogene 1; SPAG9, sperm-associated antigen 9; TSHR, thyroid-stimulating hormone receptor; CDKN2B, cyclin-dependent kinase inhibitor 2B; MAPK, mitogen-activated protein kinase; SOS1, SOS Ras/Rac guanine nucleotide exchange factor 1; KRAS, KRAS proto-oncogene; mTOR, mammalian target of rapamycin; MMP, matrix metallopeptidase; HMGA, high-mobility group A; ↔, interaction; ↑, upregulation; ↓, downregulation; →, induction.

### 4.2. LncRNAs Deregulated in ATC

#### 4.2.1. The Prader-Willi/Angelman Region RNA5 (PAR5) lncRNA

Using a microarray-based methodology, the aberrant expression of several lncRNAs in nine ATC samples versus five NT samples used as controls was found. Among the deregulated lncRNAs, 19 lncRNAs were found to be increased, and 28 were found to be decreased with fold changes of >1.1 and <−1.1, respectively, and a *p*-value of <0.05. Among the decreased group of lncRNAs, PAR5 undoubtedly is a key player in regulating the tumor progression of TCs (Figure 4A). Indeed, PAR5 levels drastically decrease in undifferentiated thyroid tumors (including ATCs) but remain stable in WDTCs (such as PTCs), suggesting that the role of PAR5 is strictly associated with cancer progression [97]. Consistently, the restoration of PAR5 expression is able to slow the proliferation and migratory ability of TC cells, explaining its association with the acquisition of a malignant phenotype [97].

Interestingly, a tumor suppressor role of PAR5 was also confirmed by studies on other cancer types. Indeed, PAR5 has also been found to be decreased in hepatocellular carcinoma and gliomas [118,119], where, interestingly, PAR5 has been reported to exert a tumor suppressor role through its interaction with EZH2. In fact, through a molecular interaction, PAR5 blocks the ability of EZH2 to alter the methylation state of histones [120], thus unblocking the expression of a series of important genes that regulate cell proliferation and the stem-cell phenotype. This mechanism of action also seems to work in ATC. In fact, PAR5 is not only able to interact with EZH2, blocking its function, but it is also able to regulate its protein stability [97], as already reported for other lncRNAs [121,122] (Figure 4A). This observation is also supported by the inverse correlation of EZH2 protein levels with those of PAR5 in ATC tissues [97] and by previous studies that reported EZH2 overexpression in ATC but not in PTC and FTC [87]. It is also worth noting that PAR5 is able to block the action of EZH2 on the E-cadherin promoter (reported to be repressed through H3K27 trimethylation exerted by EZH2 [123]) by reducing its binding to chromatin, thus allowing its expression. The role played by PAR5 is, therefore, clear in the late stages of neoplastic transformation and, in particular, during EMT and loss of cell differentiation.

EZH2 overexpression seems to represent a critical event in the development of ATCs since the deregulated expression of several ncRNAs contributes to regulating the expression and functions of EZH2, as observed for miRNAs [88,124], circRNA [125], and additional lncRNAs [126]. Indeed, in ATC, there is a decrease in miR-25 and miR-30d levels, inversely correlated with the increased levels of EZH2. Interestingly, functional experiments demonstrated that these two miRNAs are able to target EZH2 and, consequently, impair the proliferation of ATC cells in culture, resulting in their accumulation in the G2/M phase of the cell cycle [88]. These observations, on the one hand, confirm the oncogenic role played by EZH2 and, on the other hand, confirm the importance of PAR5 as an additional actor in the complex functional mechanism of EZH2.

#### 4.2.2. The Rhabdomyosarcoma 2-Associated Transcript (RMST) lncRNA

The same analysis that revealed the decrease in PAR5 also led to the identification and subsequent characterization of another lncRNA, RMST, which drastically decreased in ATCs [97]. RMST levels decrease in a gradual manner, starting from differentiated to undifferentiated TCs, linking its expression to the differentiation state, and they are associated with an increase in SRY-box transcription factor 2 (SOX2), a critical regulator of pluripotent stem cells. These results were also confirmed by an additional in silico analysis using public microarray data (De Martino, M. et al., manuscript in preparation).

Moreover, functional studies carried out following the restoration of its expression have also shown that RMST is able to reduce the oncogenic capacity of ATC cells (proliferation and migration), as well as reduce the stemness of anaplastic cancer stem cells (De Martino, M. et al., manuscript in preparation). RMST is also able to strongly regulate EMT markers by repressing mesenchymal markers and inducing epithelial ones. Interestingly, RMST expression levels strongly decrease following the induction of thyrospheres, starting from ATC cells growing in adherent conditions (De Martino, M. et al., manuscript in preparation).

Therefore, these findings support an important role of RMST downregulation in the development of ATC, which is also supported by the extremely low levels of RMST detected in triple-negative breast carcinomas (TNBC) and the ability of its restored expression to slow the proliferation and invasion of TNBC cells [127].

In conclusion, all the findings preliminary reported by the authors (De Martino, M. et al., manuscript in preparation) regarding the involvement of RMST in the regulation of the stemness properties and tumor-initiating ability of thyroid cancer stem cells (CSCs) may provide new therapeutic opportunities for patients affected by untreatable TC such as ATC.

### 4.3. LncRNAs Deregulated in Both PTC and ATC

#### 4.3.1. The Papillary Thyroid Carcinoma Susceptibility Candidate 3 (PTCSC3) lncRNA

Recently, several other lncRNAs have been investigated in TCs, looking for a novel and valuable therapeutic, diagnostic, and prognostic biomarker. The first lncRNA studied in PTC, the so-called papillary thyroid carcinoma susceptibility candidate 3 (PTCSC3), was found to be downregulated in TC [98] (Figure 5A). Its tumor suppressor role was further confirmed by functional experiments in which several TC-derived cell lines transfected with PTCSC3 showed reduced growth and an increased apoptotic rate.

Moreover, it has been reported that miR-574-5p, a previously described oncogenic miRNA, may be sponged by PTCSC3. In this manner, PTCSC3 loss could drive thyroid carcinogenesis by upregulating miR-574-5p [98] (Figure 5A). Consistently, PTCSC3 has also been described as a tumor suppressor in ATC. Indeed, it is able to inhibit signal transducer and activator of transcription 3 (STAT3) expression, which promotes tumoral progression and doxorubicin resistance, mainly by inducing the expression of the INO80 complex ATPase subunit (INO80), a DNA repair master regulator [99]. Consequently, PTCSC3 depletion leads to a reduction in doxorubicin sensitivity in ATC-derived cellular models [99] (Figure 5A). Moreover, in the PTCSC3-overexpressing cell lines, Prominin 1 (PROM1) and ATP-binding cassette subfamily B member 1 (ABCB1) expression levels, two cancer stem-cell markers, were found to be diminished [99].

#### 4.3.2. The Metastasis-Associated Lung Adenocarcinoma Transcript 1 (MALAT1) lncRNA

Another interesting lncRNA is involved in thyroid tumorigenesis. Indeed, MALAT1 lncRNA has been reported as an onco-lncRNA in various cancer types, such as hepatocarcinoma [128], colon cancer [129], and breast cancer [130]. Moreover, Liu, J. et al., found that MALAT1 expression levels were increased in PTCs compared to normal tissues, and they were correlated with the invasiveness and tumoral stage [100].

Further studies have also reported the oncogenic role of MALAT1 in medullary thyroid carcinomas (MTCs). Indeed, MALAT1 was found to be upregulated in almost all of the analyzed samples; importantly, in MTC-derived cell lines, the inhibition of MALAT1 expression led to a decrease in cell growth and invasion [101].

The oncogenic role of MALAT1 has also been confirmed in ATC [102]. In particular, MALAT1 is able to sponge the tumor suppressor miRNA miR-363-3p, derepressing the MCL1 apoptosis regulator (MCL1) oncogene, a target of miR-363-3p. Moreover, BI-847325, an MAPK/ERK kinase (MEK) and Aurora kinase family inhibitor, significantly affected MALAT1 expression and, consequently, the expression of its downstream genes, such as miR-363-3p, MCL1, and cyclin D1 (CCND1) in two ATC cell lines (C643 and SW1736), while also inducing cell cycle arrest in the G1 phase and increasing apoptosis [102] (Figure 5B). Thus, MALAT1 also acts as an oncogene in ATC, regulating the expression of cell cycle- and apoptosis-related genes.

#### 4.3.3. The Urothelial Carcinoma-Associated 1 (UCA1) lncRNA

UCA1 was first found to be upregulated in bladder cancer [131], where it mainly acts by sponging several cancer-related miRNAs. UCA1 is overexpressed in PTC samples while also correlating with tumor stage and the presence of metastases [103]. Furthermore, in PTC cell lines, UCA1 overexpression induces an increase in cell proliferation and suppresses apoptosis [103]. Li, D. et al., reported that UCA1 is able to compete with the bromodomain-containing 4 (BRD4) oncogene when binding to miR-204. Thus, UCA1 knockdown impaired PTC cell growth by decreasing BRD4 expression levels [104,105] (Figure 5C). The oncogenic role of UCA1 also seems to be related to its sponging activity in ATC. Indeed, it is able to compete with the c-MYC oncogene when binding to miR-135; thus, UCA1 upregulation derepresses c-MYC from miR-135 inhibition, inducing cell growth and proliferation [105] (Figure 5C). These findings suggest that lncRNA UCA1 is also an oncogene in TCs, representing a novel potential target for its treatment.

#### 4.3.4. The H19 Imprinted Maternally Expressed Transcript (H19) lncRNA

H19 has been described as a pivotal regulator in both embryonic development and tumorigenesis [132], and it is deeply involved in thyroid tumorigenesis. Indeed, it is overexpressed in PTC, thus inducing EMT [106]. Moreover, H19 is able to sponge miR-17-5p, thus promoting cell cycle progression via the upregulation of YES proto-oncogene 1 (YES1) [107] (Figure 5D).

In ATC, H19 silencing induced a decrease in cellular proliferation and invasion, as well as an increase in apoptotic rate [108]. Intriguingly, by using an in vivo model of ATC lung metastases, it was reported that H19 knockdown decreases metastasis formation and tumoral mass [108], highlighting this lncRNA as a potential target in ATC therapy.

#### 4.3.5. The Nuclear Paraspeckle Assembly Transcript 1 (NEAT1) lncRNA

The role of NEAT1 in thyroid carcinogenesis is validated by the reduction in cell survival, migration, and invasion of the PTC-derived cell line TPC-1 after silencing of its expression, as well as by the ability of its overexpression to significantly increase the size of PTCs obtained in xenograft mouse models [109], thus revealing its oncogenic properties. MiR-214 binds to NEAT1, whose overexpression, acting as a sponge, leads to the downregulation of miR-214 levels [109].

Moreover, NEAT1 knockdown causes a reduction in cisplatin resistance in ATC-derived cells by sponging miR-9-5p [110]. Indeed, miR-9-5p directly targets and downregulates the sperm-associated antigen 9 (SPAG9), which is responsible for the resistance to cisplatin in ATC cells [110] (Figure 5E).

### 4.4. LncRNAs Deregulated in FTC

#### 4.4.1. Landscape of lncRNAs Aberrantly Expressed in FTC

With regard to the analysis of aberrant lncRNA expression in FTC, Yakushina, V. D. et al., evaluated the expression of lncRNAs in the different subtypes of TCs; through the intersection of the genes differentially expressed in comparison to NT tissues, they identified lncRNAs common or specific to a precise subtype [133].

They found that 13 lncRNAs were differentially expressed in FTA and WDTC in comparison to NT. Interestingly, some of them (LINC02555 and LINC02471) gradually increased going from FTA to FTC and PTC, while others (ENSG00000256542 and ENSG00000258117) gradually decreased from PTC to FTC and FTA [133]. Six lncRNAs were differentially expressed in WDTC versus FTA. However, no lncRNAs were found to be differentially expressed in FTC versus FTA. Among these lncRNAs, LINC02454 showed the highest positive fold change value [133]. Interestingly, 19 lncRNAs were specific to FTC in comparison with PTC and ATC, with ENSG0000281383 being the only one overexpressed, while the others were downregulated. However, even in this case, lncRNAs were also found deregulated in FTA. Among the downregulated lncRNAs, SLC16A1-AS1 and KRT7-AS showed the strongest negative fold change values [133]. The biological processes in which these FTC-specific lncRNAs are involved appear to be associated with the mRNA splicing process. Indeed, recent findings confirm the role played by aberrant splicing events in cancer development and progression [134,135].

In conclusion, this study did not show a lncRNA expression signature that can distinguish between FTC and FTA, which would allow a differential diagnosis, confirming their common origin.

#### 4.4.2. The PVT1 Oncogene (PVT1) lncRNA

A recent study reported an important role of the PVT1 lncRNA in FTCs [111]. Indeed, this lncRNA was found to be overexpressed in TCs, including FTCs, and functional studies on cultured cell lines showed that its expression is critical in positively regulating the cell cycle and proliferation. Furthermore, a fundamental aspect of the role of PVT1 is the ability to regulate the expression of the thyroid-stimulating hormone receptor (TSHR) by interacting with the EZH2 protein (Figure 4B) [111].

#### 4.4.3. The Growth Arrest-Specific 5 (GAS5) lncRNA

Another study reported that GAS5 was downregulated in a cohort of 212 TCs, including 67 FTCs, compared with 61 benign thyroid tumors. It is interesting to note that very low levels of GAS5 are correlated with TNM, lymph node metastases, reduced overall survival, high-risk AMES, and a worse prognosis for patients [136]. Therefore, this study proposed GAS5 as a novel tumor suppressor gene.

Very recently, the molecular mechanism via which GAS5 acts in FTCs was also demonstrated [112]. Indeed, the levels of GAS5 were inversely correlated with those of miR-221-3p in FTCs; GAS5 would sequester miR-221-3p by a sponge effect, leading to an increased expression of its target, cyclin-dependent kinase inhibitor 2B (CDKN2B), which in turn blocks the cell cycle and thereby reduces the proliferation of FTC cells (Figure 4B) [112], supporting the role of GAS5 as a tumor suppressor gene in TCs.

## 5. Deregulation of Pseudogenes in TC

### 5.1. The BRAF Pseudogene 1 (BRAFP1)

The BRAFP1 pseudogene seems to be involved in thyroid carcinogenesis since it is able to transform NIH3T3 cells in vitro and induce tumors in nude mice by activating the MAPK signaling pathway; importantly, a positive correlation between the expression levels of BRAF and its pseudogene was detected in PTCs [137]. Interestingly, an inverse correlation between BRAFP1 mRNA levels and the presence of the BRAF^V600E^ mutation (enhancing the oncogenic activity of BRAF in about 40% of PTCs) was observed, suggesting that the activation of BRAFP1 or the V600E mutation is required to drive the transformation of the thyroid normal follicular cell [113].

### 5.2. The Double Homeobox A Pseudogene (DUXAP) 8 and 10

An analysis of The Cancer Genome Atlas (TCGA)-TC (THCA) data showed that the pseudogene DUXAP8 was drastically upregulated in PTC, and its overexpression was related to an advanced grade and worse prognosis in PTC patients [114]. It was also demonstrated that miR-20b-5p could directly target DUXAP8 and that DUXAP8 expression was positively associated with that of SOS Ras/Rac guanine nucleotide exchange factor 1 (SOS1), which activates the KRAS proto-oncogene (KRAS) [114]. These results unveiled the existence of a ceRNA mechanism among DUXAP8, SOS1, and miR-20b-5p (Figure 6A).

The DUXAP10 pseudogene was also found to be strongly overexpressed in PTC cells in comparison with NT cells. DUXAP10 silencing blocks cell proliferation and invasiveness, decreases matrix metallopeptidase 2 (MMP-2) and matrix metallopeptidase 9 (MMP-9) expression, and enhances the apoptotic rate and caspase 3 activity in PTC cells [115]. Moreover, the AKT/mammalian target of rapamycin (AKT/mTOR) pathway was repressed when DUXAP10 was knocked down in PTC cells, highlighting the oncogenic role of this pseudogene in PTC [115] (Figure 6B).

### 5.3. The High-Mobility Group A (HMGA) 1 Pseudogene 6 (P6) and 7 (P7)

Two HMGA1 pseudogenes, HMGA1P6 and HMGA1P7, were recently identified. They regulate the expression of HMGA1 proteins via a ceRNA mechanism. In fact, HMGA1P6 or HMGA1P7 overexpression inhibited the influence of miRNAs on both HMGA1 transcript and protein levels in a dose-dependent manner [116,117]. The HMGA proteins positively or negatively control the transcription of several genes involved in neoplastic cell transformation, and their overexpression represents a feature of human malignancies correlating with poor survival [138]. Interestingly, HMGA1P6 and HMGA1P7 also regulate the expression of EZH2, HMGA2, and other cancer-related genes with a critical role in cancer progression by sponging miRNAs able to target these genes (Figure 6C). HMGA1P6 and HMGA1P7 are drastically upregulated in ATC but not in PTC [25,116], thus contributing to cancer progression. It is worth noting that HMGA1P6 and HMGA1P7 also show oncogenic activity in vivo since transgenic mice overexpressing them developed diffuse B-cell lymphomas [27].

### 5.4. The EGF-like and EMI Domain-Containing 1 Pseudogene (EGFEM1P)

Interestingly, EGFEM1P, a putative pseudogene containing an EGF-like and an EMI domain, was found to be highly overexpressed in PTC [77]. It has been suggested that the EMI domain could represent a protein–protein interaction module, as the EMI domain of elastin microfibril interfacer (EMILIN) 1 was found to interact with the gC1q domain of EMILIN2 [139]. This pseudogene is also upregulated in lung adenocarcinomas, suggesting a role of its upregulation in carcinogenesis.

## 6. The Impact of the Identification of ncRNAs in the Clinical Management and Outcomes of TC

Several studies have already confirmed that ncRNAs may also have an important role in the clinical management of patients with TC in particular and cancer in general. In fact, it is not surprising that the evaluation of some ncRNAs selected as disease biomarkers may represent a valid tool in the diagnosis and prognosis of TC, with a predictive role in determining the response to therapy.

The first important clinical aspect of the importance of ncRNAs may be a diagnostic one, allowing the differential diagnosis of different TC histotypes. For example, the analysis of PAR5 lncRNA levels, which are drastically decreased in ATCs, but not changed in PTCs, may help in the differential diagnosis between PTC and ATC; generally, PAR5 could be used as a possible marker of aggressiveness [97]. In clinical practice, fine-needle aspiration biopsy (FNAB) is routinely used to identify benign from potentially malignant thyroid nodules. However, in a fraction of cases (~20%), the diagnostic results obtained in cytology are somewhat ambiguous and, thus, do not allow a clear differential diagnosis between benign and malignant nodules, with consequent nonoptimal administration of the thyroid patient [140]. In this perspective, the analysis of selected lncRNAs in FNABs can represent a valid support tool for the definitive diagnosis decision, and the use of molecular biomarkers is also encouraged by international guidelines [141]. Furthermore, the evaluation of lncRNAs also takes place in a fairly minimally invasive way, which suggests increasing the use of lncRNAs in the diagnosis of thyroid nodules. In addition to this, the evaluation using FNAB on a series of miRNAs differentially expressed between benign and malignant thyroid lesions has now been verified by several studies. In particular, the evaluation of the different combinations of expression of some miRNAs (miR-221, miR-222, miR-146b, and miR-181b) led to the conclusion that miRNAs have a very promising role in defining the correct diagnosis of the disease [142].

In different types of cancer, it has also been found that lncRNAs can be directly evaluated in biological fluids through the optimization of PCR-based procedures in order to detect even very low amounts in the bloodstream, as reported for MALAT1 [143,144], which represents a valid biomarker to be used in clinical practice. In fact, a strong upregulation of MALAT1 has been reported for a wide range of human cancers, while its tissue levels are also associated with the clinical characteristics of patients, including resistance to therapy, indicating it as an ideal clinical biomarker [143]. In the perspective of minimally invasive diagnosis, the evaluation of different miRNAs in the plasma and serum of patients has proven to be even more useful, thanks to their molecular nature that allows for carrying out robust investigations of miRNA blood expression in terms of reproducibility, sensitivity, and specificity [142,145].

A second crucial aspect of the possible use of ncRNAs is related to the treatment of TC. In addition to the surgical removal of the entire organ, treatments include chemotherapy regimens and the administration of radioactive iodine (RI) [146]. However, several clinical and molecular factors contribute to the treatment resistance of an important fraction of patients. In this perspective, ncRNAs can help stratify patients according to the prediction of resistance to treatment in order to optimize treatments and avoid unnecessary ones. In this regard, it would be important to evaluate the expression of the NEAT1 lncRNA in PTC, which was found to be abundantly overexpressed in cultured cells that acquired resistance to RI treatment. Furthermore, its silencing favored the restoration of sensitivity to treatment [147]. Therefore, NEAT1 could be used as a biomarker of response to treatment and as a target in the perspective of personalized medicine. It is interesting to note that many patients with advanced TC do not respond to treatments with RI as they have very low levels of the sodium–iodide symporter (NIS) protein due to an overexpression of miR-146b. Therefore, scientists are currently evaluating the possibility of reducing miR-146b levels to promote NIS expression so as to improve RI uptake with an improved treatment outcome [148]. This type of investigation clearly suggests what the applicative potential of miRNAs may be in terms of personalized therapeutic strategies for TC. Additionally, miR-125b seems to be important in TC therapy since it was shown that its expression is able to sensitize TC cells to treatment with cisplatin by activating autophagy [149].

Lastly, the evaluation of ncRNAs can certainly assume an important prognostic aspect. This is due to the fact that a series of ncRNAs, such as those described in this review, are directly implicated in the onset and progression of TC, behaving both as oncogenes and as tumor suppressor genes and modulating fundamental aspects of the neoplastic cell, such as proliferation [150], apoptosis, differentiation, and malignancy. This is the case of lncRNA H19, which is overexpressed in PTCs compared to normal tissue, whereby its elevated levels show an association with a greater severity of the disease [106]. Furthermore, its direct involvement in the regulation of EMT makes it a disease driver [106] and, as such, a potential therapeutic target to be evaluated in the future. In terms of the prognostic aspect, a fundamental role is also played by miR-146b, which is associated with a worse prognosis for patients affected by PTC [65]. In these patients, was not only an association with the clinical parameters of the disease observed but also an association with follow-up and disease-free survival time.

In conclusion, even if very few ncRNAs can be used in the clinical setting as biomarkers in the short term, many of them appear very promising for clinical use. However, in order to use them as biomarkers in clinical practice, an extensive amount of work is still required leading to the development of well-characterized and standardized clinical protocols based on the use of ncRNAs.

## 7. Comment

Unfortunately, it is almost impossible to discuss all of the lncRNAs deregulated in TC. Clearly, a multitude of lncRNAs are deregulated in TC and somehow functionally linked to its appearance and progression. To name a few, we can refer to the promoter of CDKN1A antisense DNA damage-activated RNA (PANDAR) [151], SLC26A4 antisense RNA 1 (SLC26A4-AS1) [152], and myocardial infarction-associated transcript (MIAT) [153]. However, other reviews on the role of lncRNAs in TC were recently published (e.g., [154,155]). These reviews were more focused on the clinical aspects of lncRNA deregulation and the possibility of using their detection in bodily fluids as a diagnostic and prognostic tool. Conversely, the present review includes novel lncRNAs, not included in the previous reviews, focusing on the mechanisms via which their misregulation contributes to thyroid carcinogenesis.

## 8. Conclusions

In addition to genetic and epigenetic changes, recent findings have underlined that ncRNAs can play pivotal roles in all steps of cancer progression. In particular, lncRNAs may regulate several molecular mechanisms participating in the instauration and progression of TC, e.g., by regulating the expression of genes and miRNAs involved in proliferation, metastasis, apoptosis, and by modulating the EMT of cancer stem cells. However, in order to identify new and effective biomarkers, the ncRNA regulatory network described in this review must be further confirmed in a larger number of samples. Indeed, evaluating the prognosis and treatment of TCs, particularly ATCs, with greater efficiency should represent the next step in the ncRNA field, implementing a more personalized approach to fighting thyroid neoplasms.

## Figures and Tables

**Figure 1 cancers-14-03079-f001:**
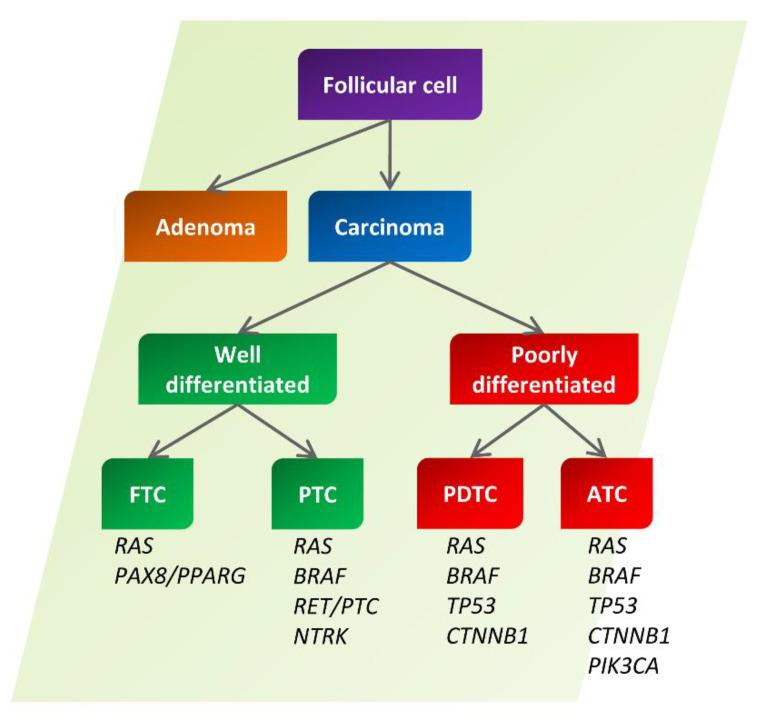
Tumors generated from human thyroid follicular cells. Schematic representation of the mainly benign and malignant tumors generated from human thyroid follicular cells. Several mutations frequently encountered in each tumor histotype are also indicated. FTC, follicular thyroid carcinoma; PTC, papillary thyroid carcinoma; PDTC, poorly differentiated thyroid carcinoma; ATC, anaplastic thyroid carcinoma.

**Figure 2 cancers-14-03079-f002:**
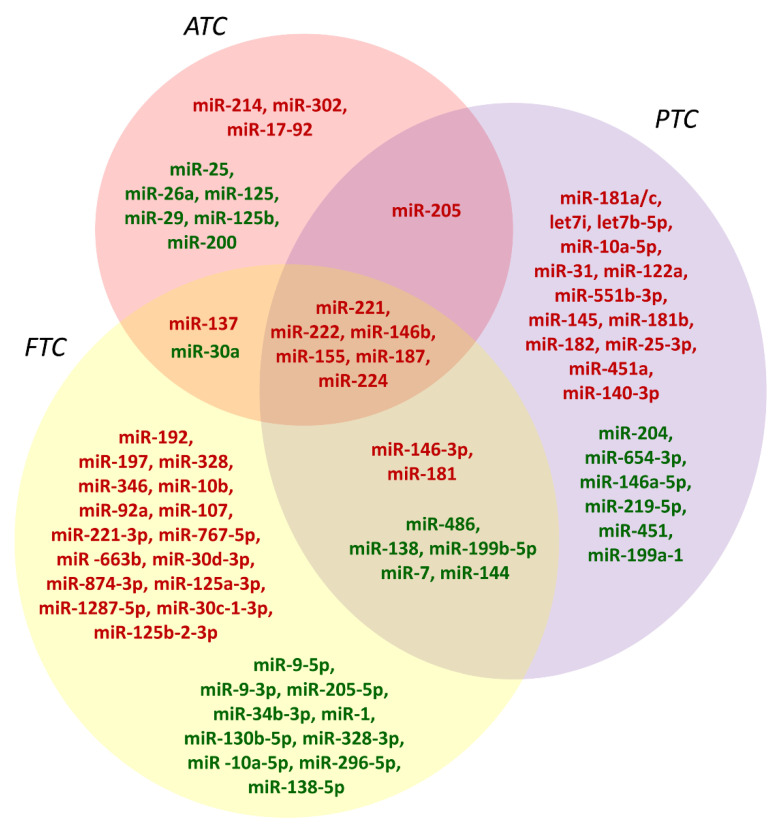
The landscape of miRNAs differentially expressed in the diverse histotypes of thyroid-follicular-cell-derived carcinomas. Upregulated miRNAs are indicated in red, while downregulated ones are indicated in green. FTC, follicular thyroid carcinoma; PTC, papillary thyroid carcinoma; ATC, anaplastic thyroid carcinoma.

**Figure 3 cancers-14-03079-f003:**
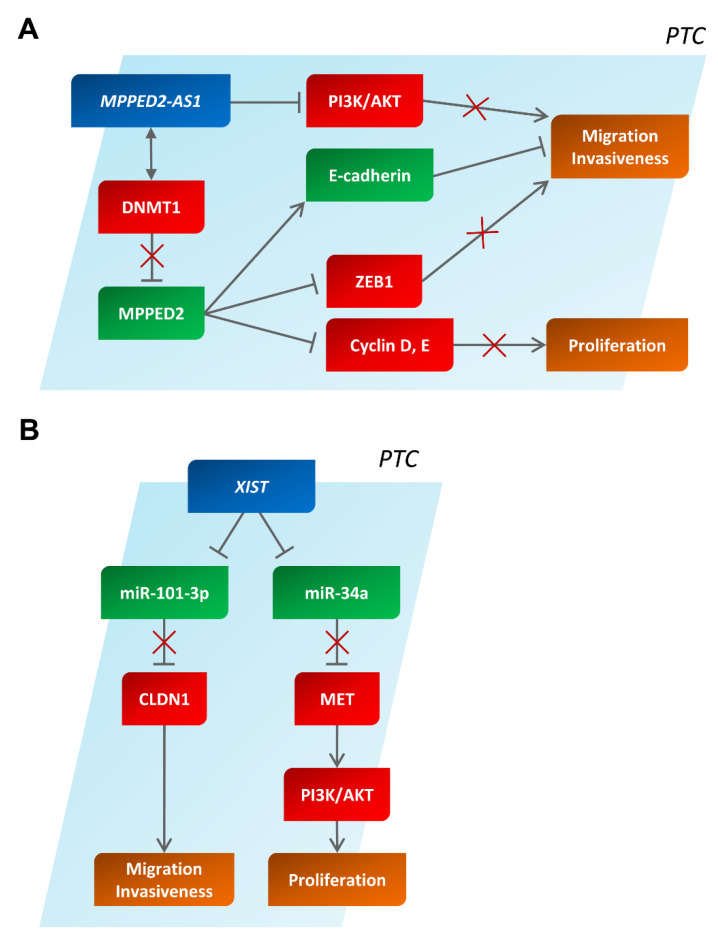
The involvement of MPPED2-AS1 and XIST in human papillary thyroid carcinoma. Schematic representation of several interactions explaining the role of MPPED2-AS1 (**A**) and XIST (**B**) in human papillary thyroid carcinomas. In red and in green, partners with an oncogenic or anti-oncogenic behavior are reported, respectively. PTC, papillary thyroid carcinoma.

**Figure 4 cancers-14-03079-f004:**
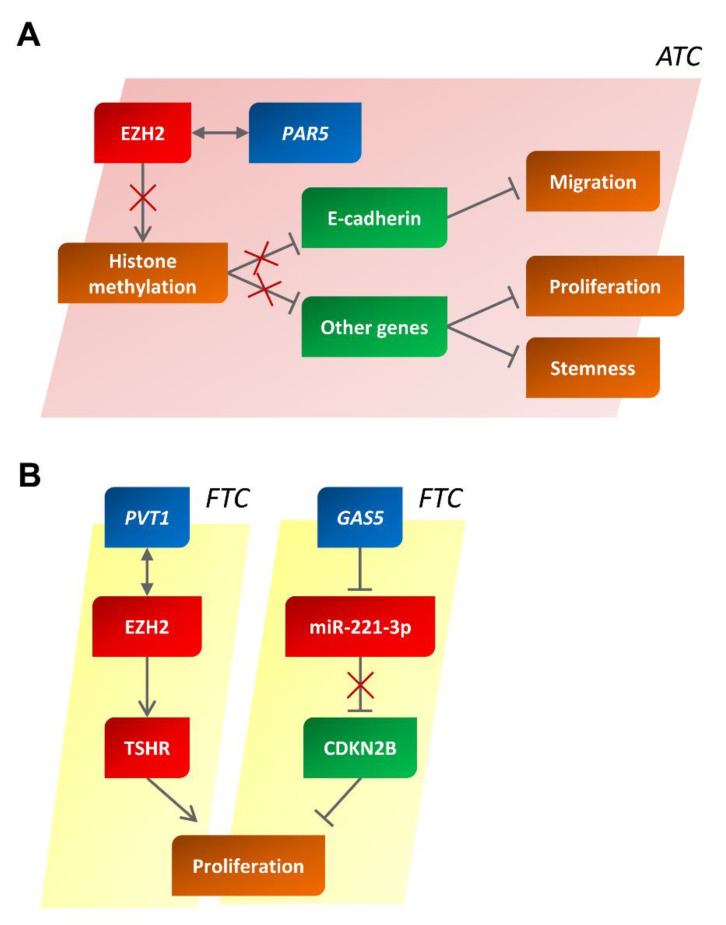
Deregulation of PAR5 and PVT1/GAS5 in anaplastic and follicular thyroid carcinomas. Schematic representation depicting the main pathways in which PAR5 (**A**) and PVT1/GAS5 (**B**) are thought to be involved. In red and in green, partners with an oncogenic or anti-oncogenic behavior are reported, respectively. ATC, anaplastic thyroid carcinoma; FTC, follicular thyroid carcinoma.

**Figure 5 cancers-14-03079-f005:**
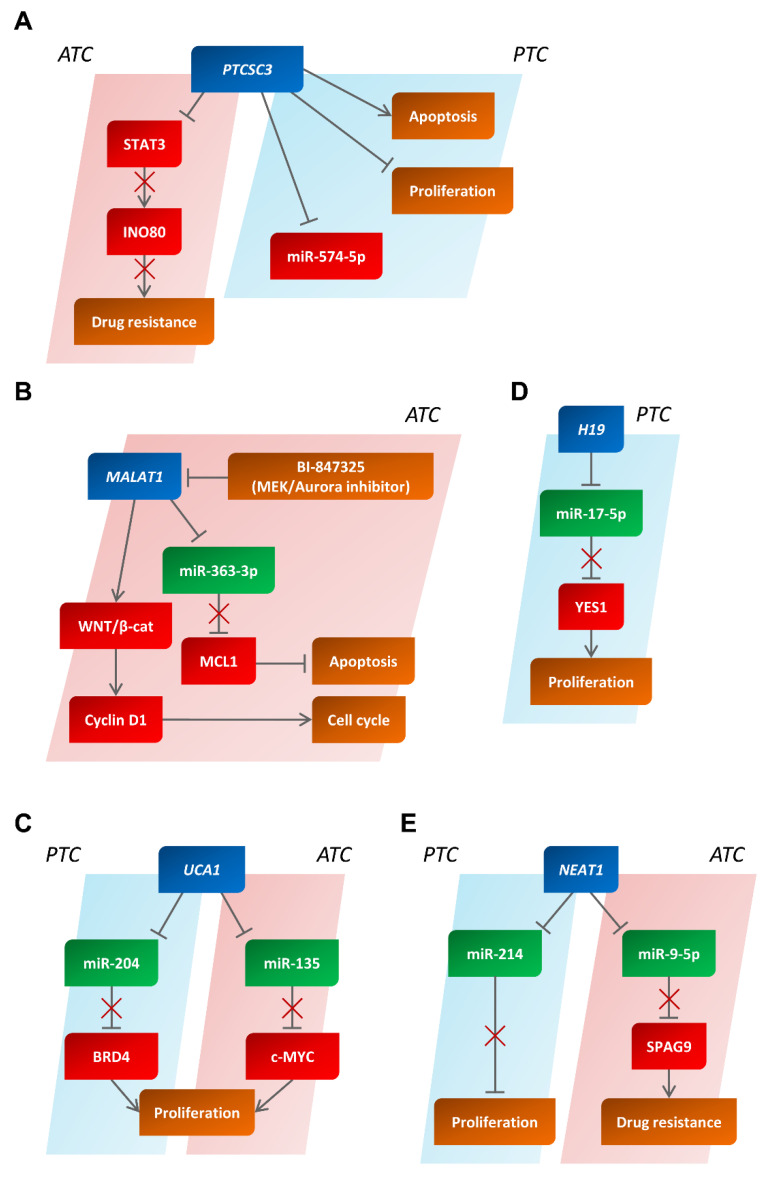
The landscape of lncRNAs deregulated in human papillary and anaplastic thyroid carcinomas. Schematic representation of the main lncRNAs involved in human papillary and anaplastic thyroid carcinomas. The suppressive role of PTCSC3 (**A**) and the oncogenic behavior of MALAT1 (**B**), UCA1 (**C**), H19 (**D**), and NEAT1 (**E**) are schematically reported in the figure. In red and in green, the partners with an oncogenic or anti-oncogenic behavior are reported, respectively. PTC, papillary thyroid carcinoma; ATC, anaplastic thyroid carcinoma.

**Figure 6 cancers-14-03079-f006:**
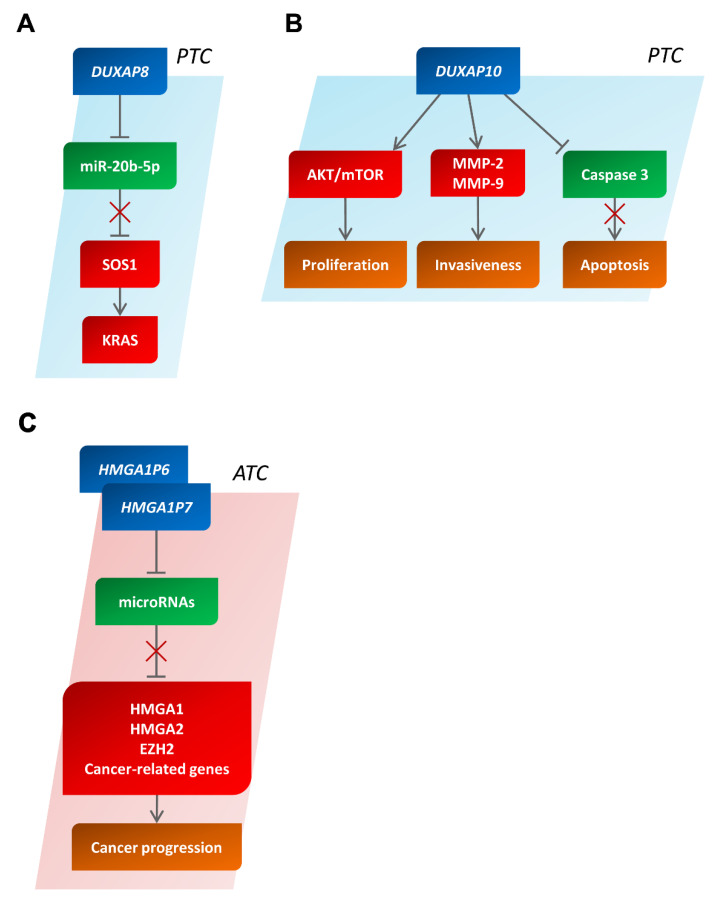
The involvement of pseudogenes in the biology of human thyroid carcinomas. Schematic representation showing the functions of DUXAP8 (**A**) and DUXAP10 (**B**) in papillary thyroid carcinomas and of HMGA1P6 and HMGA1P7 (**C**) in anaplastic thyroid carcinomas. In red and in green, the partners with an oncogenic or anti-oncogenic behavior are reported, respectively. PTC, papillary thyroid carcinoma; ATC, anaplastic thyroid carcinoma.

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
