# Peer review of "Noncoding RNAs in Thyroid-Follicular-Cell-Derived Carcinomas"

_cancers, 2022, doi:10.3390/cancers14133079_

Round 1

Reviewer 1 Report

The manuscript of a review article describing ncRNAs in thyroid cancer (TC) has been extensively revised, and I feel that it has been improved in quality. However, I have to mention that there still remain serious structural concerns in this manuscript, which makes it unfriendly for readers. This manuscript focuses on miRNAs and lncRNAs in TC; however, the structures in Section 3 (describing deregulated miRNAs in TC) and the following sections are not uniform. Concretely, Section 3 is subdivided into subsections based on TC types (e.g., PTC and ATC), while Sections 4-7 are subdivided into subsections corresponding to individual lncRNAs. This reviewer suggests to rearrange the Sections 4 and 5 as follows, for example.

4. Deregulation of lncRNAs in TC

4.1. LncRNAs deregulated in PTC

4.1.1. MPPED2 and MPPED2-AS1

4.1.2. XIST

4.1.3. HOTAIR

4.1.4. RP11-230G5.2

4.1.5. AC079630.2

4.2. LncRNAs deregulated in ATC

4.2.1. PAR5

4.2.2. RMST

4.3. LncRNAs deregulated in both PTC and ATC

4.3.1. PTCSC3

4.3.2. MALAT1

4.3.3. UCA1

4.3.4. H19

4.3.5. NEAT1

4.4. LncRNAs deregulated in FTC

4.4.1. PVT1

4.4.2. GAS5

5. Deregulation of pseudogenes in TC

5.1. BRAFP1

5.2. DUXAP8 and DUXAP10

5.3. HMGA1P6 and HMGA1P7

5.4. EGFEM1P

Other concerns:

1.      Line 63: Note that RAS mutations are also detected in FA.

2.      Lines 142-169: This reviewer suggests removing this section because circRNAs are not mentioned in the following sections.

3.      Lines 447-463: These sentences seem logically curious. Correlations of LRRK2 mutations (not expression) with Crohn's diseases etc. are not directly associated with the pathological relevance of LRRK2 in PTC. There should be more appropriate literatures describing the role of LRRK2 in TC, e.g., https://pubmed.ncbi.nlm.nih.gov/31180559/. Further, the second half of this section seems to be less associated with AC079630.2 but rather related to other ncRNAs such as miR-138-5p and H19.

4.      Lines 758-811: Not only lncRNAs but also miRNAs should be mentioned in this section.

5.      Figures 1, 3, 4, and 5: The outer frames drawn in these figures are misleading. I had an impression that these frames indicate cell membranes, but actually not.

Author Response

Reviewer 1

Comments and Suggestions for Authors

The manuscript of a review article describing ncRNAs in thyroid cancer (TC) has been extensively revised, and I feel that it has been improved in quality. However, I have to mention that there still remain serious structural concerns in this manuscript, which makes it unfriendly for readers. This manuscript focuses on miRNAs and lncRNAs in TC; however, the structures in Section 3 (describing deregulated miRNAs in TC) and the following sections are not uniform. Concretely, Section 3 is subdivided into subsections based on TC types (e.g., PTC and ATC), while Sections 4-7 are subdivided into subsections corresponding to individual lncRNAs. This reviewer suggests to rearrange the Sections 4 and 5 as follows, for example.

  1. Deregulation of lncRNAs in TC

4.1. LncRNAs deregulated in PTC

4.1.1. MPPED2 and MPPED2-AS1

4.1.2. XIST

4.1.3. HOTAIR

4.1.4. RP11-230G5.2

4.1.5. AC079630.2

4.2. LncRNAs deregulated in ATC

4.2.1. PAR5

4.2.2. RMST

4.3. LncRNAs deregulated in both PTC and ATC

4.3.1. PTCSC3

4.3.2. MALAT1

4.3.3. UCA1

4.3.4. H19

4.3.5. NEAT1

4.4. LncRNAs deregulated in FTC

4.4.1. PVT1

4.4.2. GAS5

  1. Deregulation of pseudogenes in TC

5.1. BRAFP1

5.2. DUXAP8 and DUXAP10

5.3. HMGA1P6 and HMGA1P7

5.4. EGFEM1P

We want first to thank this Reviewer for understanding our effort to improve the entire manuscript. In this perspective, in order to substantially improve manuscript reading as far as English is concerned, this revised version of our manuscript has been enhanced through the use of MDPI's English editing service to improve submitted manuscripts.

As far as the manuscript structure is concerned, we fully agree in modifying it by inserting the sub-subheadings in sections 4 and 5, according to the precious suggestion of this Reviewer.

Other concerns:

  1. Line 63: Note that RAS mutations are also detected in FA.

We sincerely apologize for failing to include this information. In the revised version of the manuscript we have corrected this point.

  1. Lines 142-169: This reviewer suggests removing this section because circRNAs are not mentioned in the following sections.

While we were quite in agreement with this Reviewer, and since we had not made any mention of circRNAs in the original version of the manuscript, we were nonetheless convinced to keep this short part (for completeness on ncRNAs) as requested in the previous round of reviews by another Reviewer.

  1. Lines 447-463: These sentences seem logically curious. Correlations of LRRK2 mutations (not expression) with Crohn's diseases etc. are not directly associated with the pathological relevance of LRRK2 in PTC. There should be more appropriate literatures describing the role of LRRK2 in TC, e.g., https://pubmed.ncbi.nlm.nih.gov/31180559/. Further, the second half of this section seems to be less associated with AC079630.2 but rather related to other ncRNAs such as miR-138-5p and H19.

We thank this Reviewer for pointing this out to us. In the new version of the manuscript we have proceeded to modify this part, inserting a short comment based on the work suggested by this Reviewer. At the end of the paragraph we have however kept a very small reference to the expression association of LRRK2 with H19 and RP11-476D10.1, as previously requested by the other Reviewer.

  1. Lines 758-811: Not only lncRNAs but also miRNAs should be mentioned in this section.

We fully agree with this Reviewer and, consequently, following his suggestion we have proceeded to insert comments regarding the microRNAs as well.

  1. Figures 1, 3, 4, and 5: The outer frames drawn in these figures are misleading. I had an impression that these frames indicate cell membranes, but actually not.

We sincerely thank the Reviewer for this comment. Since we are also interested in the readability of the manuscript, we have graphically modified the figures included in the manuscript, with the aim of not misleading the reader's attention elsewhere.

Reviewer 2 Report

The authors answered all my remarks.

Author Response

Reviewer 2

We thank this Reviewer for providing us with valuable suggestions in the previous round of reviews, which allowed us to substantially improve the review. Additionally, as previously mentioned, the latest version of the manuscript has been qualitatively improved as regards English, by the English editing service of MDPI.

Round 2

Reviewer 1 Report

The manuscript has been revised appropiately and no further revisions are required.